# Design and Implementation of a Time-Restricted Eating Intervention in a Randomized, Controlled Eating Study

**DOI:** 10.3390/nu15081978

**Published:** 2023-04-20

**Authors:** Karen White, Beiwen Wu, Scott J. Pilla, Jeanne Charleston, May Thu Thu Maw, Lawrence J. Appel, Jeanne M. Clark, Nisa M. Maruthur

**Affiliations:** 1ProHealth Clinical Research Unit, School of Medicine, Johns Hopkins University, Baltimore, MD 21207, USA; 2Division of Epidemiology, Dalla Lana School of Public Health, University of Toronto, Toronto, ON M5T 3L9, Canada; 3Department of Medicine, School of Medicine, Johns Hopkins University, Baltimore, MD 21205, USA; 4Department of Health Policy and Management, Bloomberg School of Public Health, Johns Hopkins University, Baltimore, MD 21205, USA; 5School of Nursing, Johns Hopkins University, Baltimore, MD 21205, USA; 6Department of Epidemiology, Bloomberg School of Public Health, Johns Hopkins University, Baltimore, MD 21205, USA

**Keywords:** time-restricted eating, controlled eating study, study design, nutrition interventions

## Abstract

The efficacy of time-restricted eating for weight loss has not been established, as prior studies were limited by a lack of controlled isocaloric designs. This study describes the design and implementation of interventions in a controlled eating study evaluating time-restricted eating. We designed a randomized, controlled, parallel-arm eating study comparing time-restricted eating (TRE) to a usual eating pattern (UEP) for the primary outcome of weight change. Participants were aged 21–69 years with prediabetes and obesity. TRE consumed 80% of calories by 1300 h (military time), and UEP consumed ≥ 50% of calories after 1700 h (military time). Both arms consumed identical macro- and micro-nutrients based on a healthy, palatable diet. We calculated individual calorie requirements, which were maintained throughout the intervention. The desired distribution of calories across eating windows in both arms was achieved, as were the weekly averages for macronutrients and micronutrients. We actively monitored participants and adapted diets to facilitate adherence. We provide the first report, to our knowledge, on the design and implementation of eating study interventions that isolated the effect of meal timing on weight while maintaining constant caloric intake and identical diets during the study period.

## 1. Introduction

Time-restricted eating (TRE), in which caloric intake is restricted to specific times of the day, is a promising chrononutrition intervention to address obesity and associated metabolic risk factors [1,2]. In particular, the timing of eating during the active period (daytime in humans), consistent with circadian rhythms, may improve metabolic outcomes [3,4,5,6].

Several randomized clinical trials have evaluated the cardiometabolic effects of TRE as a chrononutrition intervention in humans [7,8,9,10,11,12,13,14,15,16,17,18,19,20,21,22,23], with some suggesting the beneficial effects of TRE on weight or glucose outcomes. In line with the hypothesis that eating earlier during the active period (daylight) is better for human cardiometabolic disease, most of these trials of TRE studied early TRE: that is to say, they confined the eating window to earlier in the day and compared this to eating later in the day [8,9,10,11,12,15,17,18,19,20,21]. Regardless of the time of day focused on, prior trials of TRE were either not eating studies [12,13,14,15,16,18,19,20,21,22] or eating studies with substantial limitations including short duration of intervention [8,9,10,17], small sample size [8,10,11,23] and low retention rates [7,10,11,23], limiting the inferences that can be drawn. These limitations make it difficult to assess whether prior promising results of TRE are due to reductions in caloric intake or other confounding factors other than the timing of food intake. A critical issue is whether a benefit of TRE on weight results from reduced caloric intake; if so, then TRE is just another means to reduce calories.

Controlled diet studies, in contrast to strictly behavioral interventions, have the potential to address these issues by providing precise, controlled calorie and nutrient intake. In a controlled diet study, all the participants’ food is provided to them for the duration of the study period. This is the ideal study design to evaluate the efficacy of TRE as the same caloric intake and nutrient balance can be maintained between arms, thereby isolating the effect of meal timing on outcomes.

Our metabolic kitchen has a long history of controlled diet studies and is best known for participation in the multi-center DASH eating study [24]. Most prior controlled diet studies, including ours, have focused on evaluating the impact of dietary patterns on health outcomes. A study comparing responses to two eating patterns composed of the same menus but with different distributions of calories over the day presents a unique challenge, and detailed methods for intervention design in this setting, to our knowledge, have not been described elsewhere.

In this article, we describe the design and implementation of the interventions in a randomized, controlled, isocaloric eating study (Time-Restricted Intake of Meals (TRIM) Study) comparing the effect of early TRE (eating within a window of time earlier in the day) versus a usual eating pattern (UEP; eating later in the day) on weight and other cardiometabolic outcomes.

## 2. Materials and Methods

### 2.1. Overall Design

We conducted a randomized, parallel-arm, 12-week eating study of two dietary interventions (Figure 1): (1) TRE arm—in which participants consumed all calories between 0800 and 1800, with 80% prior to 1300; (2) UEP arm—in which participants consumed calories throughout the day (between 0800 and 0000), with at least 50% after 1700. We included adults in this study who had obesity (BMI 30–50 kg/m^2^) and prediabetes (hemoglobin A1c 5.7–6.4%) or diabetes (hemoglobin A1c 6.5–6.9%), not requiring medications. We selected this population with obesity at high risk of diabetes because of the joint public health burden of diabetes and cardiovascular disease [25] and the need to identify effective weight loss interventions for this population [26]. Also, the available evidence on TRE has suggested a favorable impact on glucose homeostasis [9,10,23,27], and those with impairment in glucose homeostasis (e.g., prediabetes) are likely to demonstrate an effect of TRE on glucose homeostasis.

Potential participants completed one week of run-in eating to test their acceptance of both timing-of-eating patterns prior to randomization. We took a constant clinical research diet approach in which we specified and calculated intake for the planned menus prior to the study using specialized software (ESHA Food Processor 11.5 (2018), ESHA Research, Salem, OR, USA); weighed controlled portions; ensured consistent food sources and constant food preparation procedures in our metabolic research kitchen (Baltimore, Maryland); and discouraged any food replacements to the extent possible [28]. We advised participants to maintain their usual level of activity throughout the intervention period.

The primary outcome for the clinical trial was body weight, and secondary outcomes were changes in fasting glucose, homeostatic model assessment for insulin resistance (HOMA-IR), area-under-the-curve for glucose from 2-h oral glucose tolerance testing (OGTT), and glycated albumin. The outcomes for this paper are the distributions of calories throughout the meal-timing windows.

The trial was registered at clinicaltrials.gov (NCT03527368).

### 2.2. Screening and Run-In

The focus of our screening procedures was to select and randomize participants who were likely to safely and successfully complete this efficacy trial. Research staff conducted a telephone screening with potential study participants, during which participants were asked about medical issues, medications, and willingness/ability to participate in the study. Participants meeting initial eligibility criteria were invited for in-person screening to determine eligibility (major inclusion criteria: age 21–69 with obesity and either prediabetes or diabetes not requiring medications; major exclusion criteria: unable to participate in interventions, sleep/circadian disorders, use of glucose-lowering or weight-affecting medications, or major medical illness). After confirmation of clinical eligibility based on medical history (via questionnaire) and physical measures (height, weight, and blood pressure) at the initial in-person visit, participants completed a dietary questionnaire at a second in-person visit, which asked about food allergies and intolerances and barriers to completing the study’s dietary interventions (Appendix A). A research dietitian reviewed this questionnaire in detail with the participant to assess dietary eligibility for the study.

After the in-person screening, participants had to successfully complete a seven-day run-in period prior to randomization. Day 1 of the run-in began with a 60-min study orientation conducted by a research dietitian, study coordinator, and the Principal Investigator. Orientation included an overview of the study details and requirements, as well as food demonstrations of meal completion expectations. During the study orientation, participants were provided samples of specific foods that were found to be problematic in prior controlled diet studies (e.g., cottage cheese, nuts and milk). Orientation hand-outs included Safe Foods to Go (Appendix A); Allowed Beverages (Appendix A); Allowed Seasonings (Appendix A); a sample food diary for monitoring adherence; and timing of meals for both study arms. After orientation was completed, participants received study food to be eaten until their next visit to the research site. During the run-in period, participants ate a meal on-site three times on three separate days and were provided food for four days of TRE (including two weekend days) and three days of UEP (see below for details on diet composition and timing of meals below). Therefore, participants had the opportunity to try all study meals and the different timing of eating interventions during the run-in period. During the run-in period, research dietitians observed participants and assessed their ability to attend on-site meals and willingness to eat all study foods during the various eating time windows covered by both interventions. This phase also provided dietitians an opportunity to get to know the participants and communicate with them regarding the importance of only going forward to randomization if they could truly commit to the 12 weeks of the study.

After the completion of the run-in, the Principal Investigator, lead research dietitian, director of recruitment and retention, and study coordinator overseeing data collection met in person to assess the final eligibility of each participant prior to randomization. During this case conference, the lead research dietitian was asked to confirm the eligibility of each participant from a dietary intervention perspective. At the time of randomization, the director of recruitment and retention again confirmed each participant’s willingness to participate in the 12-week intervention regardless of the intervention assignment.

### 2.3. Randomization

Briefly, participants were randomized 1:1 to the TRE or UEP intervention with stratification on gender. A statistician created the computer-generated random number sequence with randomly permuted blocks of 2 and 4 and placed randomization assignments in individually sealed envelopes that were sequentially numbered. Randomization assignment was given to each participant during a randomization visit by a study staff member.

### 2.4. Assessment of Daily Caloric Needs

Since the effect of timing of eating on weight was the primary study aim, we sought to determine the baseline daily caloric need for each participant prior to randomization. This calorie level was then held constant throughout the intervention period to ensure that changes in weight and other outcomes were related to the timing of eating and not changes in caloric intake. We used the Mifflin-St. Jeor equation to determine daily caloric need [29]. This method entails different equations for men and women. Variables in the equation are age, height (cm) and weight (kg). This equation provides a basal caloric need, which is multiplied by an activity factor (see below) to determine the daily caloric need.

We determined the activity level using the IPAQ-SF [30]. A research dietitian met with each participant to review their IPAQ-SF, at which time the dietitian asked the participant to further elaborate on their activities and eating habits. The dietitian then assigned an activity factor. Typical activity factors used include 1.2 for sedentary, 1.4 for moderate activity, and 1.6 for very active [31]. We selected the following activity factors for this study: 1.3 for very low/sedentary, 1.4 for low, 1.5 for moderate activity, and 1.6 for high activity (Appendix A). We selected these factors because of the younger population in our study (eligibility range: 18–69) and our experience with prior studies [32,33]. Research dietitians used clinical judgment to assign activity factors based on the IPAQ-SF and discussions with participants.

### 2.5. Diet Composition and Menu Planning

The major objective of this trial was to determine the effect of the timing of eating. Therefore, in order to reduce the effect of confounding from other aspects of diet, the composition of the TRE and UEP interventions were identical in foods and nutrient content. These dietary interventions only differed in the timing of food distribution throughout the day (Figure 1). We focused on developing a diet that would be both healthy and palatable. The nutrient composition of the diets was similar to the OMNI Heart Unsaturated Fat Diet [32] and the SPICE Study [34]. In Omni Heart [32], we modified the original DASH Diet, which was previously shown to be beneficial for blood pressure reduction [24], to understand the impact of varying macronutrients on cardiovascular disease risk factors. In Omni Heart, the diet richer in unsaturated fat, the “Omni Heart Unsaturated Fat Diet,” had beneficial effects on estimated cardiovascular risk [32]. In SPICE, the focus of the intervention was taste perception in the setting of low sodium intake, and thus a particular emphasis was placed on flavor [34].

We developed a seven-day menu cycle at five calorie levels (1600, 2000, 2500, 3000, and 3500 kcal). Table 1 shows the nutrient targets by kcal level, and Appendix A display that we achieved the desired distribution of calories across eating windows in both arms. We used the energy method to establish the ranges for micronutrients at each energy level as follows [35]. We set nutrient targets at the 2000 calorie level based on previous studies and then indexed to the other calorie levels using the nutrient density at 2000 kcal. For example, the sodium target at 2000 kcal was set to 2300 mg. Thus, the sodium density was 2300 mg/2000 kcal or 1150 mg/1000 kcal. This density was then used to calculate the target value for menus at other energy levels. For example, the sodium target at 2500 kcal was calculated as 2875 mg, which had the same sodium density as the sodium target at 2000 kcal.

Due to the percentage of calories allotted to each meal, we divided servings of many foods and recipes between meals; for example, one day, participants in both arms would have part of a serving of lentil kale bean salad at breakfast and part of a serving of lentil kale bean salad at lunch (Appendix A). We developed 100-calorie unit foods (“energy cookies”) to match the dietary nutrient targets and used these to meet calculated calorie needs that fell between the calorie levels listed above. We created two recipes for these unit foods to minimize taste fatigue. Although we did not set meal-specific macronutrient targets, we planned to achieve our nutrient goals across all meals each day.

While recipes were planned to be flavorful, we acknowledged that participants might want additional spices and herbs to use as needed on their foods. We provided participants with a list of allowed spices and herbs that were free of sodium and calories (Appendix A) and provided a minimal amount of potassium. Regarding beverages, there were some days when milk and juice were part of the planned menus. Water was encouraged as the additional beverage of choice. In order to encourage adherence, participants were provided with guidelines for allowed beverages (Appendix A). This included the allowance of one serving of alcohol, 8 ounces of plain coffee or tea, and 12 ounces of diet soda daily. Specific powdered drinks were allowed as desired; these drinks had to be free of sodium and potassium and less than 5 calories per serving. They were allowed one individual serving of nondairy coffee creamer daily.

We planned menus on a one-week cycle using the ESHA software program [version 11.4.548]. Most recipes came from previous studies, primarily the SPICE study, which consisted of recipes that were well accepted by participants [34]. The nutrients we targeted in this study include protein, total carbohydrate, total fat, saturated fat, dietary fiber, calcium, potassium, and sodium. Although many of these nutrients are required to be listed on the Nutrition Facts labels by U.S. Food and Drug Administration (FDA), there are some exceptions. To ensure we had a complete nutrient profile for each product used in study menus, research dietitians first searched for the product in ESHA software. If the product was not available in ESHA or had missing information, then research dietitians would input the nutrient profile from the Nutrition Facts label, the manufacturer’s website, and/or general USDA values, in that order.

Using the calculated menu, the research dietitians created recipes and production sheets for use by the kitchen staff. We trained kitchen staff on the accuracy of measurement for research food preparation. Next, we tested the recipes and determined cooked factors. The cooked factor [36], also known as the cooking yield and retention factor, is a critical part of the nutrient calculation for cooked foods when composition data are not available. Cooked factors are used to reflect the changes in food weights resulting from moisture and fat losses during cooking and cooling processes. Our protocol for establishing a cooked factor is shown in Appendix A based on the type of cooking method for a given recipe.

We obtained foods from a central vendor and specific grocery stores to achieve consistency of nutrients across successive cohorts and simplify the process for updating nutrients if there was a change in the product through the vendor.

### 2.6. Timing of Eating

We developed eating windows and distribution of calories within those windows (Figure 2) with the following principles in mind: (1) prioritizing consuming more calories earlier in the day for the TRE arm and later in the day for the UEP arm; (2) prioritizing testing of time-restricted eating that may promote healthy circadian rhythm rather than focusing on the fasting duration; (3) achieving high participant adherence; and (4) implementation of a usual eating pattern based on the literature [37,38]. To be consistent with these principles, we planned for 80% of calories to be consumed by 1300 in the TRE arm (with all calories consumed by 1800). In the UEP arm, at least 50% of calories were to be consumed after 1700.

### 2.7. Controlled Eating

Participants ate lunch or dinner on-site in a central dining room three weekdays each week. They were provided with study foods to take home for the other days, including weekends. Participants were asked to complete a daily diary to monitor their adherence to the study diet each day of the intervention, which included checking off the time they consumed each meal each day and reporting any study food that was not eaten or food that was eaten that was not provided by the study. At each on-site meal day, we provided participants with a tray of food for the meal, which was fully cooked and only required microwave reheating. Participants were observed by a monitor to be sure they fully completed their meal. They were asked to notify the dietitians when they finished their meals. If any food was left on the tray or in the container by a participant, the participant would be asked to finish the meal, and reasons for not completing the meal were noted if the participant was unable to complete the meal. If any food was dropped on the table or on the floor, the food was weighed by a dietitian and replaced with the same food if possible; otherwise, the food and the quantity of food would be recorded on the daily diary form.

### 2.8. Staffing

We dedicated significant effort to the onboarding and training of study staff and implemented specific staffing patterns to support the safety and success of the timing-of-eating intervention. We required all kitchen staff to obtain and maintain food safety certification (ServSafe^®^, Chicago, IL, USA). With this background in place, we trained each kitchen staff member on the standardized preparation of study recipes; this included learning and preparing each recipe using specific cooking techniques and measurements with the observation by a dietitian. A study dietitian was in the kitchen at all times to observe and ensure the safety and standardized preparation of study meals. Staffing patterns consisted of a morning shift and evening shift to ensure sufficient staffing in the preparation of the metabolic kitchen at the start of each day and the close-out of the kitchen at the end of each day.

### 2.9. Methods for Adherence and Retention

A full description of methods for assessing and optimizing adherence and adherence results is provided in a separate report [39]. Briefly, we asked participants to self-report adherence each day of the study period using a standardized paper form. Dietitians were available to answer questions, clarify aspects of diet and timing, review daily diaries during the on-site eating and assist with problem-solving around adherence to the intervention. The principal investigator and clinic director also ate regularly with participants in the study dining room. This helped to establish rapport and encourage participants to adhere to the intervention. We did not exclude participants because of lactose intolerance; the study provided Lactaid^®^ (lactase enzyme supplement) and Beano^®^ (alpha-galactosidase supplement) for participants as needed. For participants who had raised intolerances as a possible concern (e.g., lactose intolerance), the dietitians spent extra time confirming that the lactase enzyme supplementation was sufficient. To encourage adherence to the intervention during holidays, we prepared special menus for those days that were close in nutrient content to the regular menu but included foods that were commonly eaten on those holidays.

This study was approved by the Johns Hopkins University School of Medicine Institutional Review Board (IRB00155640), and all participants provided informed consent. Participants did receive financial remuneration (up to $325) for participation in the study.

### 2.10. Sample Size

We performed an *a priori* sample size estimate for the primary outcome: At the time of protocol development, evidence from a prior study [12] suggested an expected effect size of −2 kg relative to control (standard deviation of 2 kg within arm). With a two-sided alpha of 0.05, we estimated 17 participants per arm for 80% power. Based on prior studies, we anticipated 90% completion, so estimated a need for 20 participants within each arm (for a total of 40 participants for the study).

## 3. Discussion

In this publication, we describe the design of a time-restricted eating intervention that addressed many of the challenges of prior studies of time-restricted eating. We developed a palatable, acceptable, healthy diet with a timing of eating windows that were feasible for a 12-week intervention. We provided all food for participants throughout the study and required them to eat on-site for three meals each week. This controlled setting provides scientific validity without requiring an inpatient stay for the entire study. In total, these features enabled this study to deliver valid results on the true impact of time-restricted eating on cardiometabolic outcomes.

This report provides important details on the procedures underlying the successful implementation of the study interventions, including building flexibility into the intervention (e.g., offering substitutions when possible) and devoting extra resources to retention, such as special planning for the conduct of the intervention during holidays. Operational details important to our success included the consolidation of food acquisition from a few sources and assuring that staff members were well-trained in cooking techniques and food safety, which provided for a smooth food preparation flow.

While our research dietitians and staff at this facility were experienced in planning research diets [24,32,33,34,40], the design of the menus to accommodate different distributions of calories throughout the day provided a new challenge. First, we had to consider what time windows for the intervention and comparison groups would allow us to answer our scientific questions while also being feasible. We considered work schedules and typical times of food intake in the US [37,38] to select windows. The next challenge was to provide the exact same menu of foods to both arms: the TRE group had large breakfasts and lunches, and small dinners, while the opposite was true for the comparison groups. In some cases, breakfast and lunch foods were not traditional (e.g., participants received kale bean salad for breakfast on Thursdays), and the different timing windows also posed an additional logistical challenge for intervention staff when packing food to be eaten off-site.

Given the resources that are necessary to recruit participants willing and able to participate in a controlled diet study (i.e., consume only study food for the entire study period), consideration of palatability and some flexibility in intervention delivery is important. In this trial, participants were required to eat the same food with the same timing of eating pattern for 12 weeks. Also, there were participants who found it difficult to eat a large amount of food in one meal, and some participants reported that they could not exercise as usual due to being too full after the larger meals. Special attention was paid to utilizing prior study recipes that were diverse and had been met with the greatest acceptance to encourage participant adherence for this lengthy study. The eating patterns required time manipulations for many participants, such as needing to arrange their study-designated mealtimes around their work schedule and church service times. We also made accommodations to the study foods when it was determined that the changes would not significantly affect nutrient targets.

Estimating participants’ caloric needs was a difficult task in this study. In most controlled diet studies testing dietary patterns, weight is held constant; participants are typically weighed at each clinic visit, and calories are adjusted to maintain weight. The primary outcome of this study was weight, and we sought to keep calories constant while only varying the timing of eating to understand how timing affects weight. Under- or over-estimation of caloric needs could result in increased hunger or difficulty eating all study food, respectively. Nonetheless, if there was an error in estimating calorie intake, the error would be random and would unlikely bias the results of our trial.

Additional potential limitations of this study are the selection of eating windows and the distribution of calories within those windows. We selected these to optimize having a contrast between intervention arms consistent with our study question while also seeking to have both interventions be feasible. While our choice of windows and distribution of calories was informed by the literature, multiple windows and distributions of calories within those windows could be considered, and we decided to evaluate one particular set of eating windows and calorie distributions in this study. The findings of this study will need to be interpreted as such, as they will only apply to early TRE with a focus on TRE during the daytime and will not necessarily be applicable to all TRE interventions.

While this study represents one of the longest controlled diet studies with a primary outcome of weight and effects of weight should be apparent by 12 weeks, this study is still relatively short-term and will not address the impact of TRE long-term. Longer efficacy studies, necessitating considerable resources, would be needed to understand the true long-term effect of TRE in humans.

Finally, implementation of these interventions took considerable resources and funding, which would be difficult for free-living persons. However, this study, as implemented, was intended to be an efficacy study and, therefore, provide evidence on if TRE is effective for weight loss in the setting of isocaloric intake. Application of this efficacy evidence in a real-world setting will require additional investigation, including a focus on understanding the mechanism of effect and a focus on implementation methods for fidelity and scaling in larger and longer behavioral intervention studies.

## 4. Conclusions

We provide the first report, to our knowledge, on how to design and implement research diets in a controlled diet study to test the isolated effect of timing of eating on weight and other cardiometabolic outcomes. Our intervention design addresses many of the challenges of prior studies and should inform future studies in this area.

## Figures and Tables

**Figure 1 nutrients-15-01978-f001:**
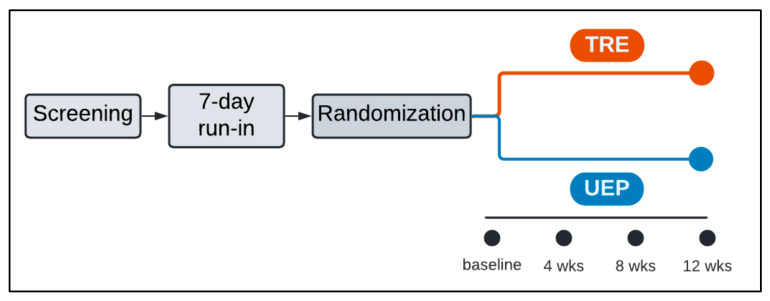
Study design. After screening for eligibility, participants completed a one-week run-in during which they experienced both meal-timing patterns. After successful completion of the run-in, participants were randomized to either the 12-week TRE (time-restricted eating) or UEP (usual eating pattern), during which they consumed only study food according to their assigned meal-timing pattern. During the 12-week intervention, participants consumed three meals per week on-site.

**Figure 2 nutrients-15-01978-f002:**
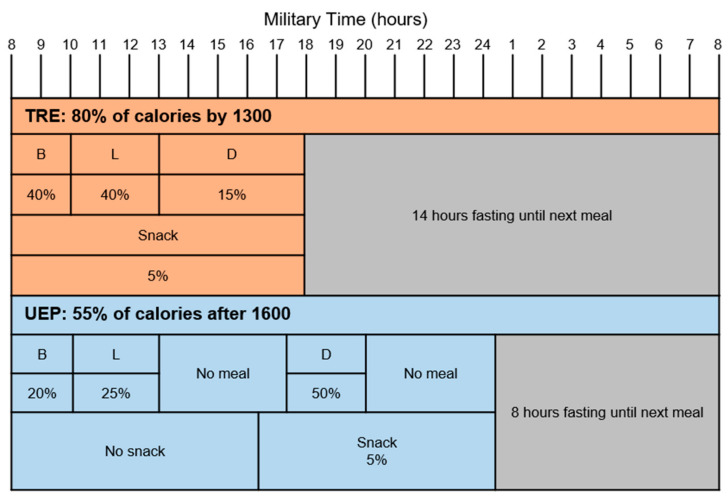
Eating windows and distribution of calories throughout the day. In the TRE arm, participants consumed all calories between 0800 and 1800 and fasted for 14 h beginning at 1800; participants consumed 40%, 40%, and 15% of total daily calories at breakfast, lunch and dinner, respectively. In the UEP arm, participants consumed all calories between 0800 and 0000 and fasted for 8 h beginning at 2000; participants consumed 20%, 25%, and 50% of total calories at breakfast, lunch, and dinner, respectively.

**Table 1 nutrients-15-01978-t001:** Nutrient Targets by Calorie Level.

	1600 kcal		2000 kcal		2500 kcal		3000 kcal		3500 kcal	
Diet Component	Target	Mean (SD)	Target	Mean (SD)	Target	Mean (SD)	Target	Mean (SD)	Target	Mean (SD)
Calories, kcal	1600	1616.2 (13.4)	2000	2010.3 (18.5)	2500	2507.2 (20.3)	3000	3001.4 (30.7)	3500	3499.0 (9.3)
Protein, %kcal	15–18	17.3 (1.0)	15–18	16.6 (0.4)	15–18	17.7 (1.2)	15–18	16.8 (1.0)	15–18	17.2 (0.6)
Carbohydrate, %kcal	45–50	45.3 (1.2)	45–50	46.1 (1.4)	45–50	45.1 (1.3)	45–50	46.1 (1.3)	45–50	46.5 (1.3)
Fat, %kcal	32–37	37.3 (0.9)	32–37	37.3 (1.4)	32–37	37.2 (1.7)	32–37	37.0 (0.9)	32–37	36.3 (1.2)
Saturated	<10	7.0 (1.6)	<10	6.7 (1.3)	<10	6.9 (1.7)	<10	6.8 (1.6)	<10	6.8 (1.3)
Calcium, mg/d	560–800	764.8 (137.1)	700–1000	896.1 (135.7)	875–1250	1200.6 (210.2)	1050–1500	1392.0 (222.6)	1225–1750	1513.8 (251.8)
Potassium, mg/d	2000–2800	2637.3 (137.2)	2500–3500	3179.9 (203.4)	3125–4375	4015.2 (167.4)	3750–5250	4686.9 (290.5)	4375–6125	5509.4 (350.9)
Sodium, mg/d	1840	1854.4 (19.2)	2300	2241.8 (34.7)	2875	2711.2 (92.4)	3450	3146.3 (93.0)	4025	3889.3 (143.7)
Fiber, g/d	>20	23.9 (3.7)	>25	29.9 (3.7)	>30	36.1 (4.6)	>38	43.9 (6.8)	>44	49.5 (5.6)

## Data Availability

The data presented in this study are available on request from the corresponding author. The data are not publicly available due to privacy reasons.

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
