# Peer review of "Design and Implementation of a Time-Restricted Eating Intervention in a Randomized, Controlled Eating Study"

_nutrients, 2023, doi:10.3390/nu15081978_

Round 1

Reviewer 1 Report (New Reviewer)

The manuscript written by White et al. demonstrates the design and implementation of a time-restricted feeding intervention in a RCT. A subset of them have already been published, but this additional information is important and valuable because the study design was special. The manuscript is well-written and easy to read. This reviewer has no comment.

Author Response

Thank you for taking the time to review our manuscript.

Reviewer 2 Report (New Reviewer)

The paper presents a protocol design and implementation of the TRIM study, a 12-week diet controlled TRF intervention.

Trying to differentiate effect of calorie restriction from meal timing is of interest. However, there would be nothing wrong if TRE resulted in decreased calorie intake, which it probably does as it results from minimal ~3% weight loss in most studies.

Limitations of past studies are well outlined. Short term is one of them. However, this proposed 12 week intervention is relatively short.

Would use ‘eating’ rather than ‘feeding’ for a human study.

Please define ‘feeding study’ Line 45. The choice of term is not the best. What about ‘controlled diet studies’?

Methods

Study design:

The TRF intervention manipulates 2 variables: restricting the eating window (8:00-18:00) as well as moving 80% of calorie intake prior to 13:00, compared to 50% of calories consumed > 17:00 in the UFP group (8:00-0:00).

It is unclear how this cut off (13:00 and 17:00) were decided upon and why the distribution of calories (80% and 50%) differ between groups.

What were the timing of meals during the one week  run in period?

L134: participants tried all study meals during the one week run-in period. How varied was the diet offered over 12 weeks?

How was/will adherence to the eating window monitored in ambulatory settting?

Was any monetary compensation allocated to the participants?

Please provide a power analysis for each outcome variables.

This is an interesting protocol to test the efficacy of a 12 week TRE intervention under eucaloric conditions. Missing are the power analysis, the rationale for the calorie distribution in the 2 groups, an additional level of intervention that may muddy the effect of time restriction alone, and the lack of long term follow up. Mechanisms are not discussed and or assessed. Translational applicability will also be limited.

Author Response

We appreciate your review of our manuscript. Please see attachment for our responses.

Round 2

Reviewer 2 Report (New Reviewer)

Some corrections have bene made, Fig 1 mentions UFP rather than UEP.

The power analysis is now presented. It is possible that UEP group will also lose weight under controlled diet.

Author Response

Thank you for your careful review of our manuscript.

Reviewer comment: Some corrections have bene made, Fig 1 mentions UFP rather than UEP.

Author response: We have removed the version of Fig 1 that mentions "UFP."

Reviewer comment: The power analysis is now presented. It is possible that UEP group will also lose weight under controlled diet.

Author response: Yes, we agree that is possible, and having a control group should help us to account for the impact of the change in diet versus the change in timing of eating. 

This manuscript is a resubmission of an earlier submission. The following is a list of the peer review reports and author responses from that submission.

Round 1

Reviewer 1 Report

White and al. described in the present article the design and implementation of a randomized, controlled isocaloric feeding study, comparing the effect of time restricted feeding versus a usual feeding pattern on weight and other cardiometabolic outcomes.

The present work is interesting and some points are described extensively and in great detail, but unfortunately some very important points are also not available. These points are listed below.

The type of article is not clear! The present manuscript is a study protocol with a very very short results-part. What is meant with 'implementation' in the title of the manuscript?  Please make a clear decision whether the manuscript describes the study protocol or the completed study (with all results). 

 Line 77: Please describe the primary and secondary outcomes in detail (body weight instead of weight) and measurements of glucose metabolism!  Please match the outcomes definition of line 77 with definition in line 61. Other cardiovascular outcomes is not the same than glucose metabolism.

How were body weight and measurements of glucose metabolism measured? Please fill in this important information to the methods-part in detail!

How many on-site visits are there in this 12 weeks? Please insert a study overview figure for better understanding.

Line 68: Pre-diabetes and type 2 diabetes should be defined in detail, please.

Please add following important information to the methods-part:

- Sample size calculation

- Statistical methods

- Data management and data monitoring

- Randomization! How was the randomization done?

 Line 260: Please add the institutional review board number of the trial

The Results-Section is very confusing! 

76 participants completed dietary screening, 10 were excluded - there should be 66 participants left, but in line 265 only 45 participants completed screening and started run-in and 41 completed run-in and were randomized?!  If this is the report of a completed trial please include a study flow chart with a detailed description of all excluded participants. If this manuscript reports a study protocol please exclude the results-section.

In Line 277 White and al. write 'all 41 participants completed final data collection' - so why not the completed study results where presented?

Reviewer 2 Report

Very interesting study design and implementation developed; however it feels like the manuscript was written in a rush. Minor corrections presented below:

Line 26 - ‘All randomized participants completed the study’, information repeated, already mentioned in line 24.

Line 35 – remove ‘but unproven’

In the introduction refer to chrononutrition and expand a bit more the introduction in order to provide a more in depth discussion of the evidence available in the field.

Line 68 – justify the population choice

Line 71 – what specialized software was used? Need to add details

Line 73 – state location of the metabolic research kitchen

Figure 1 – in text you mentioned UPF arm consumed 50% of calories after 1700 but in the figure is showing 55% calories after 1600, also in text the eating window is 0800 to 0000 and in the figure legent is 0800 to 2000 – please clarify

Line 140 and 141 – please use consistent spelling for dietitian, you used both ‘dietician’ and ‘dietitian’

Line 166 – need to define the eligibility criteria (inclusion and exclusion criteria).

Line 260 – mention the research ethics committee that approved the study (if not possible need to be justified)

Line 287 – how many were randomised to each group?

Line 330 – need to add as a limitation funding and resources and comment on whether feasible in the free-living

Reviewer 3 Report

Design and implementation of a time-restricted feeding intervention in a randomized, controlled feeding study

Karen White 1, Beiwen Wu 1, Scott J. Pilla 2, 3, Jeanne Charleston 1, 2, 5, May Thu Thu Maw 2, Lawrence Appel 2 ,4, 4 Jeanne M. Clark 2, 4 and Nisa M. Maruthur 2, 4, 5 *

I am very delighted to read this detailed article regarding the design and implementation of a controlled, randomized TRE study. It will prove to be a vital resource for other groups planning to perform similar dietary intervention studies.

Specific comments:

1)    Line 162: What is the rationale behind the recommendation to eat >50% calories after 1700 in the UFP arm? The data from Gill et al, Cell Metabolism 2015 show that people eat only 43% of calories after 1700 and only 37.5% after 1800. This recommendation can artificially force participants to eat more later in the evening than their usual time.

2)    All 41 participants who were randomized, completed the study, this could be the result of an extensive screening process applied before inclusion, given only 41 out of 76 individuals screened initially completed the study. While this might cause a bias by including only highly motivated individuals in the study, it also shows the utility of an extensive screening process for the successful completion of the study. However, keeping this participant inclusion bias in mind, the dropout/adherence rate should be interpreted with caution.

3)    Given that screening of motivated participants is important to successfully complete the study, it would be very helpful for the readers if the authors could include the screening criteria and/ or scoring system used to screen the participants.

1.     Gill, S., & Panda, S. (2015). A smartphone app reveals erratic diurnal eating patterns in humans that can be modulated for health benefits. Cell Metabolism, 22(5). https://doi.org/10.1016/j.cmet.2015.09.005

Round 2

Reviewer 1 Report

Many thanks for your corrections. Nevertheless, I am sorry, but I still do not understand background and motivation of this manuscript.